# Prevalence and Variability of *Helicobacter pylori* Clarithromycin Resistance Mutations in Pediatric Patients in Poland: A Genotypic Analysis Using the Bosphore Genotyping Kit

**DOI:** 10.3390/antibiotics14040352

**Published:** 2025-03-31

**Authors:** Tomasz Bogiel, Anna Szaflarska-Popławska, Agnieszka Krawczyk

**Affiliations:** 1Department of Propaedeutics of Medicine and Infection Prevention Ludwik Rydygier Collegium Medicum in Bydgoszcz, Nicolaus Copernicus University and Clinical Microbiology Laboratory, Dr. Antoni Jurasz University Hospital No. 1 in Bydgoszcz, 9 Maria Skłodowska-Curie Street, 85-094 Bydgoszcz, Poland; 2Department of Pediatric Endoscopy and Gastrointestinal Function Testing Ludwik Rydygier Collegium Medicum in Bydgoszcz, Nicolaus Copernicus University, 85-094 Bydgoszcz, Poland; anna.szaflarska@cm.umk.pl; 3Department of Molecular Medical Microbiology, Chair of Microbiology, Jagiellonian Univeristy Medical College, 31-007 Cracow, Poland

**Keywords:** *Helicobacter pylori*, clarithromycin resistance, genotypic analysis, multiplex PCR

## Abstract

**Background:*** Helicobacter pylori* is a Gram-negative bacterium responsible for various gastrointestinal diseases, including peptic ulcers and gastric cancer. Despite available antibiotic therapies, increasing resistance to clarithromycin—a key antibiotic in eradication regimens—poses a significant challenge. This resistance is primarily linked to point mutations in the 23S rRNA gene, particularly A2143G, A2142G, and A2142C, which hinder clarithromycin binding, reducing its bacteriostatic efficacy. This study aimed to assess the prevalence and variability of clarithromycin resistance mutations in pediatric patients from Bydgoszcz, Poland. **Methods:** A total of 45 gastric biopsy samples from pediatric patients were analyzed using the Bosphore^®^ *Helicobacter pylori* Genotyping Kit v1 to detect clarithromycin resistance-associated mutations. **Results:** Among the 45 tested samples, 30 were classified as wild-type, while 12 contained resistance-associated mutations. The most frequently detected mutation was A2143G (58.3%), followed by A2142G (33.3%). One sample exhibited both A2142G and A2143G mutations, and another contained a mixture of wild-type and mutant strains. The A2142C mutation was not detected in any sample. **Conclusions:** Our findings confirm the predominance of A2143G among clarithromycin-resistant *H. pylori* strains, consistent with global trends. The detection of both mutant and wild-type strains in a single patient highlights potential co-infections or subpopulations with varying resistance profiles. Continuous surveillance and improved diagnostic tools are crucial for optimizing treatment strategies. Tailored eradication protocols based on resistance profiling are necessary to enhance treatment efficacy and mitigate the spread of resistant strains. Further research is needed to understand the clinical implications of mixed infections and double mutations in *H. pylori* resistance development.

## 1. Introduction

*Helicobacter pylori* is a Gram-negative bacterium that colonizes the gastric epithelium in humans, causing one of the most common infections in the world [1,2]. This microorganism is responsible for several diseases, including peptic ulcers, gastric mucosa-associated lymphoid tissue lymphoma, gastric cancer and chronic gastritis [1,3]. Although the majority of patients infected with *H. pylori* remain asymptomatic, and only a small percentage will develop peptic ulcers or gastric cancer after prolonged infection, it remains a leading cause of high mortality and morbidity globally.

Commonly used antimicrobials for *H. pylori* eradication include clarithromycin, metronidazole, amoxicillin, levofloxacin and tetracycline [4,5]. Over the past few decades, the rate of antimicrobial resistance has been steadily rising, which constitutes a serious threat to human health. The mechanisms of antibiotic resistance are diverse and include both changes in the cellular structures of the bacteria and enzymatic mechanisms that neutralize the drug’s activity.

Clarithromycin is a macrolide that works by inhibiting bacterial protein synthesis. *H. pylori* can develop resistance to this drug due to mutations in the 23S rRNA gene, which is the target of clarithromycin [6]. Point mutations in the V domain of 23S ribosomal RNA region can alter the binding affinity of clarithromycin for the peptidyltransferase loop, resulting in resistance to the drug. In addition, changes in transport systems can reduce intracellular drug concentrations [6,7]. The most frequent mutations in the 23S rRNA gene linked to clarithromycin resistance are A2143G, A2142G, and A2142C [7,8,9]. The sensitivity of *H. pylori* to clarithromycin is also influenced by outer membrane proteins. Proteins like HopT (BabB), HofC, and OMP31 are absent in clarithromycin-sensitive strains but were found in clarithromycin-resistant *H. pylori* strains [6,7]. Clarithromycin is one of the key antibiotics commonly included in therapy regimens for *H. pylori* infections. Resistance to macrolides, like clarithromycin, has been observed at varying frequencies (ranging from 1% to 10%) across different countries, and this resistance significantly contributes to the failure of *H. pylori* treatment protocols [10].

The primary cause of metronidazole resistance is mutations in the *rdxA* gene, which encodes oxygen-insensitive nitroreductase, and in the *frxA* gene, which encodes flavin oxidase reductase. These mutations decrease the ability of metronidazole to convert into its active forms (NO^2−^, NO_2_^2−^), which are responsible for damaging the bacterial DNA structure, leading to its ineffectiveness [6,11].

Amoxicillin, as a beta-lactam antibiotic, binds strongly to penicillin-binding proteins (PBPs) and inhibits cell wall synthesis, leading to bacterial lysis [6,12]. To date, the expression of several PBPs has been identified in *H. pylori*. The most frequent mechanism responsible for amoxicillin resistance is the presence of point mutations in the *pbp1A* gene. Moreover, mutations in the *pbp2*, *pbp3*, *hefC*, *hopC*, and *hofH* genes have also been associated with *H. pylori* resistance to amoxicillin. Additionally, the bacterium can develop resistance to amoxicillin by producing beta-lactamases, enzymes that break down the drug’s structure, rendering it ineffective. In addition, changes in the porin channels in the bacterial membrane can reduce the uptake of amoxicillin into the bacterial cell [6,12].

Levofloxacin is a fluoroquinolone antibiotic which exerts its antibacterial effects by interacting with DNA gyrase, is encoded by the *gyrA* and *gyrB* genes. Point mutations in the quinolone resistance-determining regions of *gyrA* can hinder this process, leading to fluoroquinolone resistance in *H. pylori* [8,13]. The most prevalent mutations in levofloxacin-resistant strains occur at positions 87, 88, 91, and 97 of *gyrA* [13,14,15]. Additionally, a mutation in *gyrB* at position 463 may also play a role in fluoroquinolone resistance in *H. pylori* [14].

Tetracycline, which is also used in the treatment of *H. pylori* infections, exhibits resistance mechanisms related to active drug efflux from the bacterial cell. Bacteria may produce transport proteins that pump tetracycline out of the cell, decreasing its effectiveness. Genes encoding these transporters are the primary mechanism of resistance [16,17]. In addition, resistance is largely related due to point mutations in the *tet-1* gene within 16S rRNA. The frequency of AGA (926–928) mutations is closely associated with the level of tetracycline resistance. Single- and double-base-pair mutations typically confer low-level resistance, while triple-base-pair mutations in the AGA (926–928) region of 16S rDNA are linked to high-level resistance [18,19].

As a result of prolonged antibiotic treatment, *H. pylori* can develop resistance to multiple drugs simultaneously. This phenomenon, known as multidrug resistance (MDR), makes the treatment of *H. pylori* infections more difficult, as standard treatment regimens may become ineffective. Due to the increasing resistance of *H. pylori* to antibiotics, there is a need for continuous monitoring of treatment effectiveness and the development of new therapeutic strategies. Combination therapies, along with the use of novel antibacterial agents, currently represent one of the main approaches in combating resistant *H. pylori* strains.

In light of the above, the aim of our study was to analyze the prevalence and variability of mutations associated with *H. pylori* resistance to clarithromycin among pediatric patients in Poland using Bosphore^®^ *Helicobacter pylori* Genotyping Kit v1.

## 2. Results

The majority of the tested samples are wild-type (WT); 30/45, Figure 1. Genetic material of *H. pylori* was not detected in two samples (2/45), Figure 1. Point mutations were detected in a total of 12 samples, of which one showed a double mutation (both A2142G and A2143G), and one contained both the wild type of *H. pylori* and the A2142G mutation; Figure 1.

The most common mutation among the examined gastric biopsies was A2143G (7/45). The A2142G mutation was observed almost half as frequently (3/45), Figure 2.

The A2142G mutation was observed with almost half the frequency (3/45), Figure 2. Detailed results, including histopathological evaluation, are presented in Appendix A.

To assess the association between *H. pylori* strain type (WT or with mutation) and both the intensity of inflammation activity grade and colonization density grade, statistical analysis was performed. Additionally, the relationship between strain type and the Ct value obtained during qPCR was analyzed, as Ct is inversely related to the amount of DNA present in the sample. No significant association was observed between the presence/absence of mutations and inflammation grade or colonization density grade (*p* = 0.868; *p* = 0.818, respectively). However, statistical analysis revealed that wild-type strains (without mutations) had a higher Ct value (which indicates a lower amount of DNA present in the sample) compared to mutated strains (*p* = 0.014).

## 3. Discussion

The A2142G, A2142C, and A2143G mutations in the *23S rRNA* gene are major genetic mechanisms of *H. pylori* resistance to clarithromycin, significantly affecting treatment efficacy. Monitoring these mutations is critical for optimizing eradication therapies and limiting the spread of resistance in bacterial populations. Clarithromycin, a macrolide antibiotic, works by binding to the bacterial 50S ribosomal subunit, thereby inhibiting translation and blocking protein synthesis. The mutations listed above alter the structure of the clarithromycin-binding site within the 50S ribosomal subunit. As a result, clarithromycin is unable to bind effectively, leading to the loss of its bacteriostatic activity.

The distribution of clarithromycin-resistance mutations shows regional differences. In Japan and China, A2143G mutations account for over 90% and 100% of clarithromycin resistance cases, respectively, although these findings are based on a relatively small number of patients. Additionally, the majority of clarithromycin-resistant cases in South Korea have been confirmed to involve the A2143G mutation [20,21,22].

In the United States, the prevalence of the A2142G and A2143G mutations ranges from 48% to 53% and 39% to 45%, respectively, while the A2142C mutation is observed in 0% to 7% of cases. Similarly, in Europe, the A2142G mutation is present in 23% to 33% of *H. pylori* strains’ DNA, A2143G appears in 44% to 67%, and the A2142C mutation is reported in 2% to 10% of cases.

In our study, the frequency of A2142C, A2142G and A2143G mutations in *H. pylori* was investigated and detected in tissue biopsy samples from pediatric patients hospitalized in Bydgoszcz, Poland. Our research showed that the most frequently detected mutation was A2143G (58.3%; Figure 2). This mutation occurred almost twice as often as A2142G (33.3%; Figure 2). The A2142C mutation was not detected in any case. These observations are consistent with other reports. Studies indicate that the A2143G mutation is the most prevalent in clarithromycin-resistant *H. pylori* strains, while the A2142G mutation is less common. Furthermore, the A2142C mutation is the least frequently observed among the mutations linked to *H. pylori* resistance to clarithromycin. For example, one study found that the A2143G mutation was detected in 69.8% of cases, while the A2142G mutation was present in a smaller percentage (11.7%) [23]. In turn, in a Brazilian study, the A2143G mutation was detected in 82.7% of resistant strains, while A2142G appeared in 11.5% of cases.

It is worth noting that the A2142G mutation was associated with higher minimum inhibitory concentration (MIC) values, indicating stronger resistance [24]. Similarly, a Polish study revealed that the A2143G mutation was present in 72% of clarithromycin-resistant strains, while A2142G was identified in only 9% [25]. Another study focusing on pediatric patients, both symptomatic and asymptomatic, with *H. pylori* infection also showed that the A2143G mutation was the most commonly detected. A2143G mutations were noticed in 10.9% of children and A2142G in 6.9%. The A2143G mutation was most common in the age group 5-18 years (39.1%) [26].

The results of the study conducted by researchers from Sudan are quite interesting [27]. Using allele-specific PCR, the researchers demonstrated that the A2142G point mutation was present in 9 out of 53 (~17%) samples, while the A2143G mutation was not detected in any sample using this method. Different results were obtained through DNA sequencing, which detected the A2142G mutation in one sample and the A2143G mutation in five samples. The above study showed a higher frequency (36%) of mutation detection using DNA sequencing compared to allele-specific PCR (17%) [27]. These differences indicate a lower sensitivity of allele-specific PCR compared to DNA sequencing, highlighting the importance of developing and standardizing new diagnostic methods for this purpose. To support this thesis, the study conducted by Ghaith et al. can be outlined [28]. The researchers did not detect the A2143G point mutation in any of the samples tested using PCR, despite trying different PCR protocols. However, both the A2142G and A2143G mutations were identified using DNA sequencing. The researchers concluded that there is only one nucleotide difference between the wild-type DNA and the point mutation in the DNA sequence. Therefore, unusual mutations, which are present in a large excess of wild-type alleles, are difficult to detect using traditional gene variability assays. Dietary patterns, trends in antibiotic use, and health care policies all play a significant role in the development of antibiotic resistance in a given society. Studies indicate a positive correlation between macrolide and quinolone consumption in the community and corresponding *H. pylori* resistance in European countries [29]. Furthermore, a comprehensive meta-analysis conducted by Kuo et al. revealed that the significant increase in clarithromycin resistance from 7% before 2000 to 21% in 2011–2015 in the Asia-Pacific region could be attributed to the increased consumption of macrolides [30]. Metronidazole resistance tended to be higher in developing countries such as Nepal, Bangladesh, Pakistan, Bhutan, Vietnam, and India, whereas it appeared to be lower in nations with higher socioeconomic development indices, such as Japan. The frequent use of this inexpensive antibiotic for parasitic infections, pelvic inflammatory disease, and dental infections in developing countries contributed to these differences [29,31]. A substantial rise in levofloxacin resistance was also observed, increasing from 2% before 2000 to 27% in 2011–2015. Studies by Megraud et al. [32] and Liou et al. [33] demonstrated that fluoroquinolone resistance correlated with fluoroquinolone consumption in Europe and Taiwan, respectively. Additionally, dietary factors such as high consumption of fermented foods or probiotic yogurts, which influence gastric microbiota composition, may also play a role in modulating bacterial adaptation or survival and be associated with the development of resistance or increased/reduced risk of infection [34].

An interesting case is the double mutation, which in our study was found in one sample only. The simultaneous occurrence of A2142G and A2143G mutations in the same strain or double infection is extremely rare. Similar findings were reported by researchers from Korea. Noh et al., in their analysis of 70 specimens, identified a double mutation in just one sample. Notably, the minimum inhibitory concentration of clarithromycin in this sample was the highest of all and exceeded 128 mg/L, indicating a high level of resistance in the double mutant [20]. Other studies also confirm the rarity of double mutations. For example, in a study involving 91 samples, only one sample among those resistant to clarithromycin contained a double-mutant strain, accounting for 6.2% [22]. In turn, in a large-scale study involving 431 patients, including 91 with a confirmed clinically significant clarithromycin resistance mutation, only three samples contained the double mutation A2143G and A2142G [35].

Another interesting observation is the detection of both the wild-type and mutant strains in the same patient sample, which was noted in our study in one case. We have not investigated the specific mechanisms underlying this phenomenon. However, two main possibilities should be considered—reinfection with a different strain or co-colonization by both wild-type and mutant strains. Reinfection refers to the acquisition of a new strain of *H. pylori*, possibly with different resistance patterns, after the initial infection has been treated or resolved. Co-colonization occurs when multiple strains exist simultaneously within the same host. In the case of *H. pylori*, this could lead to complex interactions between wild-type and resistant strains, potentially influencing treatment outcomes and the development of resistance. Previous studies [36] have shown that co-colonization with different *H. pylori* strains may have important implications for the persistence of infection and the success of eradication therapy and should be considered when interpreting the results of antimicrobial susceptibility tests. It has been suggested that co-colonization could lead to competitive interactions between strains, potentially influencing the overall resistance profile of the bacterial population within the host. Such dynamics may contribute to the failure of standard treatment regimens and force a more personalized approach to therapy. Distinguishing between reinfection and co-colonization can be challenging, especially without detailed molecular analysis. Techniques such as strain typing and next-generation sequencing or RAPD fingerprinting could help clarify these mechanisms and provide more precise insights into the role of strain diversity in resistance development. We will consider their use in future studies to obtain a complete picture of the molecular mechanisms responsible for resistance. There is limited data about the mixture of mutant and wild-type strains published in the literature. This is a very rare phenomenon, documented only sporadically. A study conducted by Krashias et al. showed that among 41 samples studied, only one contained a mixture of the A2142G point mutant and non-mutated strains [37]. The studies conducted by the Vietnamese-Italian research team [38] are particularly noteworthy, as a mixture of wild-type strains and mutants was found in 50% of the samples containing A2142G, which is a very high percentage. The high rate and frequency of occurrence of a mixture of mutants and wild-type strains may be age-dependent, as researchers have shown a correlation between age and the presence of the A2143G mutation. The average age of patients with a pure A2143G mutation was significantly higher than the average age of patients with a mixture of A2143G mutant strains and wild-type *H. pylori*. This may be partially explained by the long history of antibiotic use in the older group, which resulted in an exceptionally high mutation rate [38]. Similar findings have been reported in studies conducted in France (25% for A2142G; 18.6% for A2143G) [39] and in Portugal (32.3% overall) [40]. The average age of patients with only the A2143G mutation was significantly higher compared to those with a mixture of A2143G mutants and wild-type *H. pylori* strains. Additional research on the mixture of wild-type and mutant strains, especially regarding its clinical implications in *H. pylori* treatment, are needed.

In our study, two samples were negative for *H. pylori* DNA despite positive histopathology and gastroscopy results, while one sample gave an “invalid” result. These discrepancies highlight the potential challenges of molecular diagnostics. One possible reason for the failure to detect *H. pylori* DNA is a low bacterial load in the sample, which may affect the sensitivity of the genotyping test. In addition, the presence of PCR inhibitors such as mucus, blood components, or other substances may interfere with the amplification process, leading to false negative results. Another factor to consider is the inherent methodological differences between diagnostic techniques: while histopathology and gastroscopy identify *H. pylori* at the tissue level, molecular tests rely on the presence of bacterial DNA, which cannot always be adequately extracted or detected. We cannot exclude that the mentioned factors caused an error during DNA isolation of the genetic material.

The “invalid” result observed in one sample, despite repeated testing, raises concerns about potential technical issues; however, similar factors as in the case of a negative result may also contribute to this outcome, such as an insufficient amount of genetic material or the presence of substances inhibiting the PCR. Such incidents highlight the importance of proper sample collection, handling and processing to minimize potential errors. These results emphasize the need for a comprehensive diagnostic approach to detect *H. pylori*, especially in pediatric patients. While molecular tests provide valuable information on resistance mutations, detection should be based on different tests to verify the obtained results. It seems that combining genotypic methods with traditional histopathological and endoscopic techniques can increase diagnostic accuracy and improve patient management. In cases where molecular tests give equivocal or negative results despite clinical suspicion, consideration should be given to repeating the test on another sample or using an alternative accessible diagnostic method.

Interestingly, statistical analysis showed that Ct values obtained for samples with the presence of mutated *H. pylori* strains were lower, which is equivalent to a higher amount of *H. pylori* DNA in the sample compared to wild-type strains. This difference in Ct values may be explained by several factors. Mutated strains may show increased colonization ability or increased resistance to treatment, which could lead to a higher bacterial load in the sample. Resistance to certain antibiotics enables them to colonize and multiply in the stomach environment and potentially increase their abundance in the sample, thus contributing to a higher overall *H. pylori* DNA concentration. Moreover, genetic changes in *H. pylori* could impact its pathogenicity, leading to a more intense infection. This increased pathogenicity could result in a higher bacterial load in the sample, which would be reflected by a lower Ct value. Therefore, the observed difference in Ct values between the WT and mutant strains may provide valuable insights into the underlying infection dynamics and the bacterial load associated with different *H. pylori* genotypes. The lack of statistically significant associations between *H. pylori* strain type (WT or mutant) and both inflammation grade and colonization density grade suggests that the presence of mutations associated with antibiotic resistance does not significantly impact the histopathological outcome. This indicates that factors such as the host’s immune response, bacterial virulence mechanisms unrelated to antibiotic resistance, infection duration, and environmental factors have a greater influence on gastric mucosal alterations.

A significant clinical concern associated with the increasing prevalence of *H. pylori* resistance to clarithromycin is the potential for treatment failure. As resistance rates rise, standard therapy may no longer be effective, leading to persistent infections and complications such as peptic ulcers and gastric cancer. In such cases, tailored therapy approaches, including the use of alternative antibiotics or combination therapies, are essential to improve treatment outcomes. In metagenomic analysis, Ma et al. [41] showed that in first-line treatment, tailored therapy proved to be more effective than empirical therapy. In second-line therapy, no significant differences were observed between the two approaches. A similar outcome was found in treatments involving a combination of second- and third-line therapies. Regarding adverse events, no major differences were noted between the two treatment options. In the subgroup analysis of various empirical approaches, tailored therapy demonstrated the most significant benefit compared to the conventional triple therapy. In turn, in study conducted by Kim et al. [42] demonstrated that tailored therapy based on genotypic resistance was significantly more effective than empirical therapy for *H. pylori* eradication. The eradication rates were 65.7% (201/306) in the empirical therapy group and 81.9% (235/287) in the tailored therapy group. There was no difference in compliance between the two groups. In a meta-analysis conducted by Rokkas et al. [43], the researchers showed that although tailored therapy proved to be more effective than empirical treatment, a >90% eradication rate was achieved in only 15 (44%) of the studies, and >95% in only 6 (17.6%). Based on this, the researchers concluded that while tailored therapy was superior to empirical treatment, the lack of therapy optimization prevented the achievement of high cure rates (>90%). These results highlight that *H. pylori* infection, like other infectious diseases, should follow the principles of antimicrobial stewardship in line with treatment guidelines

In summary, *Helicobacter pylori* remains a significant global health concern, particularly due to its role in the development of various gastrointestinal diseases such as peptic ulcers, gastric cancer, and chronic gastritis. The increasing antibiotic resistance in *H. pylori* strains, particularly to clarithromycin, poses a major challenge for effective treatment. The resistance mechanisms, primarily linked to mutations in the *23S rRNA* gene, have been well-documented, with the A2143G mutation being the most prevalent. Our study on pediatric patients in Bydgoszcz, Poland revealed a high prevalence of the A2143G mutation, consistent with the previous findings from other global studies. Moreover, the detection of both wild-type and mutant strains in a single sample highlights the complexity of *H. pylori* resistance and the need for more advanced diagnostic tools. The presence of double mutations, although rare, further complicates treatment outcomes and underscores the importance of tailored therapeutic strategies. Additionally, the correlation between age and the presence of specific mutations suggests that the dynamics of antibiotic resistance may evolve over time, particularly in populations with a long history of antibiotic use. Given these findings, continuous surveillance of *H. pylori* resistance patterns is essential for optimizing eradication therapies and developing more effective treatment regimens to combat this persistent pathogen.

## 4. Materials and Methods

### 4.1. Sample Collection

The tests were carried out on gastric biopsy samples obtained from pediatric patients aged 2 to 17 years (*n* = 45) that were clinically identified as positive on the basis of gastroscopic/histopathological examination (according to the Sydney classification) and/or a positive result in a urease test. These patients were assessed at the Department of Pediatric Endoscopy and Gastrointestinal Function Testing at the Ludwik Rydygier Collegium Medicum in Bydgoszcz Nicolaus Copernicus University in Toruń, Poland. The inclusion criteria for the study were as follows: initial diagnosis of chronic or recurrent non-functional dyspeptic symptoms and written parental (for children < 16 years) or both parental and patient consent to upper gastrointestinal endoscopy. The research protocol received approval from the Institutional Review Board of Ludwik Rydygier Collegium Medicum in Bydgoszcz, Nicolaus Copernicus University in Toruń. Ethics committee approval code: KB 772/2018, 20 November 2018.

### 4.2. DNA Extraction

Biopsy samples intended for DNA isolation were initially subjected to mechanical homogenization for approximately one minute using manual homogenizers (Squisher-Single, ZymoResearch, Irvine, CA, USA) compatible with 1.5 mL Eppendorf-like tubes. Following thorough crushing and homogenization, the samples underwent digestion at 37 °C for 30 min in 200 µL of a trypsin solution (5 mg/mL, Trypsin EDTA solution, Sigma, Darmstadt, Germany) to enhance DNA extraction efficiency. Next, DNA was isolated using the GeneProof Pathogen Free DNA Isolation Kit (GeneProof, Brno, Czech Republic), following the manufacturer’s protocol for clinical specimens. The extracted DNA samples were subsequently stored at −20 °C until further analysis.

The DNA isolated from a reference strain (DSM 21031 *Helicobacter pylori*) served as positive control for the whole evaluation.

### 4.3. Bosphore Helicobacter pylori Genotyping Kit v1

The Bosphore^®^ *Helicobacter pylori* Genotyping Kit v1 utilizes multiplex PCR, incorporating an internal control to monitor potential PCR inhibition. In a single reaction, both *H. pylori* DNA and the internal control are simultaneously amplified using sequence-specific primers. Fluorescent signals from the amplification of wild-type and mutant *H. pylori* templates are detected via probes labeled with FAM and Cy5 at the 3′ end, using the corresponding detection channels. Meanwhile, internal control amplification produces a fluorescent signal detected by a separate probe labeled at the 5′ end with HEX, which is observed through the appropriate channel.

There were three separate reaction mixtures, each containing 12.5 μL of PCR mix; 3 μL of detection mix no. 1, 2 or 3; 0.5 μL of internal control; and 9 μL of the examined DNA. Detection mix 1 included forward and reverse primers, along with dual-labeled probes, designed to specifically target the *H. pylori* clarithromycin resistance mutation A2142G, the wild-type variant, and the internal control. Detection mix 2 comprised forward and reverse primers as well as dual-labeled probes that specifically detect the *H. pylori* clarithromycin resistance mutation A2143G, the double mutation (A2142G and A2143G), and the internal control. Detection mix 3 included forward and reverse primers along with dual-labeled probes that are specific for the *H. pylori* clarithromycin resistance mutation A2142C, the double mutation (A2142C and A2143G), and the internal control.

DNA extracted from the *H. pylori* reference strain (DSM 21031) and molecular biology-grade water (EURx, Poland) served, respectively, as positive and negative controls for the research.

The parameters of the thermal program are presented in Table 1.

A cobas z480 analyzer (Roche, Basel, Switzerland) was used to conduct the amplification reaction.

### 4.4. Data Interpretation

The interpretation of the results was performed according to the manufacturer’s recommendations [44] as shown in Table 2.

Statistical analysis was performed using IBM SPSS Statistics version 28. A *p*-value < 0.05 was considered statistically significant. Comparisons between groups for quantitative variables were conducted using the independent samples *t*-test. Relationships between categorical and ordinal variables were assessed using the chi-square test with Fisher’s correction.

## 5. Conclusions

The increasing resistance of *H. pylori* to clarithromycin remains a significant challenge in the effective treatment of infections, particularly in pediatric patients. Our study confirms that the A2143G mutation is the most prevalent resistance-associated genetic alteration, followed by A2142G, while A2142C was not detected. The presence of mixed infections, including both wild-type and mutant strains within the same patient, suggests a complex dynamic of bacterial adaptation and resistance development. Additionally, the rare occurrence of double mutations underlines the potential for highly resistant strains that could significantly impact eradication therapy. These findings highlight the need for continuous molecular surveillance and individualized treatment strategies to improve patient outcomes. Future research should explore alternative therapeutic options, such as novel antibiotics or combination therapies, to address the growing challenge of antibiotic resistance in *H. pylori* infections.

## Figures and Tables

**Figure 1 antibiotics-14-00352-f001:**
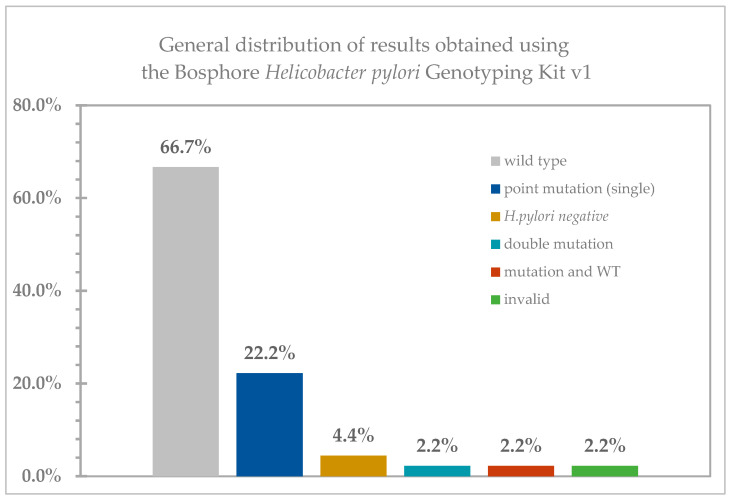
The general distribution of results in the analyzed samples (*n* = 45).

**Figure 2 antibiotics-14-00352-f002:**
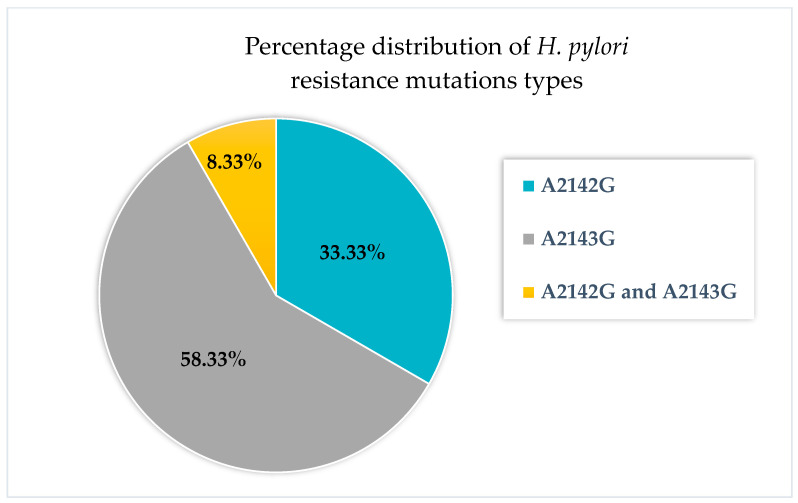
Percentage distribution of mutation types in the tested samples (*n* = 45).

**Table 1 antibiotics-14-00352-t001:** Parameters of the applied thermal program.

Step	Temperature	Time	Repeats
Initial denaturation	95°C	14:30 min.	N/A
Denaturation	97°C	00:30 min.	50 cycles
Annealing and synthesis	63°C	00:45 min.	
Hold	22°C	05:00 min.	N/A

**Table 2 antibiotics-14-00352-t002:** Possible results and interpretation criteria of the applied Bosphore® *Helicobacter pylori* Genotyping Kit v1.

Channel/Detection Mix No.	FAM	HEX (Internal Control)	Cy5	Interpretation
1	+	−	−	Clarithromycin resistance mutation 2142 A->G
1	−	−	+	Wild type
1	−	−	−	Test must be repeated
2	+	−	−	Clarithromycin resistance mutation 2143 A->G
2	−	−	+	Double mutation (2142 A->G & 2143 A->G)
2	−	−	−	Test must be repeated
3	+	−	−	Double mutation (2142 A->C & 2143 A->G)
3	−	−	+	Clarithromycin resistance mutation 2142 A->C
3	−	−	−	Test must be repeated

“+” positive result; “−“negative result.

## Data Availability

The data presented in this study are available upon request from the corresponding author.

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
