# Peer review of "Prevalence and Variability of *Helicobacter pylori* Clarithromycin Resistance Mutations in Pediatric Patients in Poland: A Genotypic Analysis Using the Bosphore Genotyping Kit"

_antibiotics, 2025, doi:10.3390/antibiotics14040352_

Round 1

Reviewer 1 Report

Comments and Suggestions for Authors

The manuscript presents a well-structured and relevant study on the prevalence and genetic variability of Helicobacter pylori (H. pylori) resistance mutations in pediatric patients in Poland. The study is timely and essential given the rising antibiotic resistance rates worldwide, particularly among pediatric populations. The methodology, results, and discussion align with the study objectives, and the manuscript is generally well-written. However, some areas require improvement, including methodological details, data presentation, and discussion clarity.

The PCR methodology should include details on quality control measures, including positive and negative controls. 

Why were Helicobacter pylori cultures and Etest assays not performed?

Table 1 would benefit from more explicit labeling and improved formatting. Currently, the presentation of patient data is confusing.

Figures 1 and 2 need improved clarity and captions that better interpret the data.

The manuscript does not specify statistical tests used to analyze the prevalence differences among the mutations. Were any statistical comparisons performed?

Incorporating confidence intervals for prevalence rates would enhance the robustness and reliability of the reported findings. The discussion describes regional variations in mutations associated with resistance but does not examine potential factors underlying these differences. Variables such as dietary patterns, antibiotic consumption trends, and healthcare policies may shape resistance rates and warrant further investigation.

The study suggests co-infection with wild-type and mutant strains occurs but does not explore potential mechanisms (e.g., reinfection vs. co-colonization).

A brief mention of potential clinical implications for treatment failure and alternative antibiotic strategies (e.g., tailored therapy approaches) would strengthen the discussion.

Comments on the Quality of English Language

No comments

Author Response

Dear Reviewer,

We thank you for your valuable comments to our manuscript. We carefully considered your comments and revised the paper according your suggestions. We responded to your comments below.

  1. The PCR methodology should include details on quality control measures, including positive and negative controls.

Thank you for your valuable comment. We fully agree that the details of quality control measures are a key aspect of the fidelity of the study results. Of course, we applied all necessary controls, therefore the mentioned aspects were included in the study and described in the current version of the manuscript in lines 374-375 and 396-398. Below, we attach the text introduced in the given lines:

„The DNA isolated from reference strain (DSM 21031 Helicobacter pylori) served as positive control for the whole evaluation”

„DNA extracted from H. pylori reference strain (DSM 21031) and molecular biology grade water (EURx, Poland) served, respectively, as positive and negative controls for the research”

  1. Why were Helicobacter pylori cultures and Etest assays not performed?

We fully agree that adding AST results for H. pylori, as recommended by EUCAST, would complete the overall picture of the study and the obtained result. Unfortunately, despite our best efforts, we were not even able to culture bacteria from the samples. Therefore, at a certain stage of the study, we gave up trying to culture bacteria and discontinued our intention to evaluate AST, focusing on the molecular evaluation of the presented study results.

  1. Table 1 would benefit from more explicit labeling and improved formatting. Currently, the presentation of patient data is confusing.

This has been corrected and the table moved to supplementary material.

  1. Figures 1 and 2 need improved clarity and captions that better interpret the data.

This has been corrected.

  1. The manuscript does not specify statistical tests used to analyze the prevalence differences among the mutations. Were any statistical comparisons performed?

Dear Reviewer, we would like to thank you for your valuable comment. Due to the relatively low number of positive samples, we decided to forgo a sophisticated statistical analysis to avoid further misunderstandings regarding the generalization of our results. However, taking into account your suggestion, we have added the appropriate sections on statistical analysis (in lines 131-139 and 413-417) and we have included the obtained results in the discussion (in lines 290-308).

„To assess the association between H. pylori strain type (WT or with mutation) and both the intensity of inflammation activity grade and colonization density grade, statistical analysis was performed. Additionally, the relationship between strain type and the Ct value obtained during qPCR was analyzed, as Ct is inversely related to the amount of DNA present in the sample. No significant association was observed between the presence/absence of mutations and inflammation grade or colonization density grade (p = 0.868; p = 0.818, respectively). However, statistical analysis revealed that wild-type strains (without mutations) had a higher Ct value (which indicates a lower amount of DNA present in the sample) compared to mutated strains (p = 0.014)”

„The statistical analysis was performed using IBM SPSS Statistics version 28. A p-value < 0.05 was considered statistically significant. Comparisons between groups for quantitative variables were conducted using the independent samples t-test. Relationships between categorical or ordinal variables were assessed using the chi-square test with Fisher's correction”

„Interestingly, statistical analysis showed that Ct values obtained for the samples with mutated H. pylori strains presence were lower, which is equivalent to a higher amount of H. pylori DNA in the sample compared to wild-type strains. This difference in Ct values may be explained by several factors. Mutated strains may show increased colonization ability or increased resistance to treatment, which could lead to a higher bacterial load in the sample. Resistance to certain antibiotics, enabling them to colonize and multiply in the stomach environment and potentially increase their abundance in the sample, thus contributing to a higher overall H. pylori DNA concentration. Moreover, genetic changes in H. pylori could impact its pathogenicity, leading to a more intense infection. This increased pathogenicity could result in a higher bacterial load in the sample, which would be reflected by a lower Ct value. Therefore, the observed difference in Ct values between the WT and mutant strains may provide valuable insights into the underlying infection dynamics and the bacterial load associated with different H. pylori genotypes. The lack of statistically significant associations between H. pylori strain type (WT or mutant) and both inflammation grade and colonization density grade suggests that the presence of mutations associated with antibiotic resistance does not significantly impact on the histopathological outcome. This indicate that factors such as the host's immune response, bacterial virulence mechanisms unrelated to antibiotic resistance, infection duration, and environmental factors have a greater influence on gastric mucosal alterations”

  1. Incorporating confidence intervals for prevalence rates would enhance the robustness and reliability of the reported findings. The discussion describes regional variations in mutations associated with resistance but does not examine potential factors underlying these differences. Variables such as dietary patterns, antibiotic consumption trends, and healthcare policies may shape resistance rates and warrant further investigation.

We agree that examining potential factors underlying regional differences in resistance mutations, such as dietary patterns, antibiotic consumption trends, and healthcare policies, are crucial. We have added a section in the discussion focus on these issues to enhance the discussion (in lines: 195-213):

„Dietary patterns, trends in antibiotic use, and health care policies all play a significant role in the development of antibiotic resistance in a given society. Studies indicate a positive correlation between macrolide and quinolone consumption in the community and corresponding H. pylori resistance in European countries. Furthermore, a comprehensive meta-analysis conducted by Kuo et al. revealed that the significant increase in clarithromycin resistance from 7% before 2000 to 21% in 2011–2015 in the Asia-Pacific region could be attributed to the increased consumption of macrolides. Metronidazole resistance tended to be higher in developing countries such as Nepal, Bangladesh, Pakistan, Bhutan, Vietnam, and India, whereas it appeared to be lower in nations with higher socioeconomic development indices, such as Japan. The frequent use of this inexpensive antibiotic for parasitic infections, pelvic inflammatory disease, and dental infections in developing countries contributed to these differences. A substantial rise in levofloxacin resistance was also observed, increasing from 2% before 2000 to 27% in 2011–2015. Studies by Megraud et al. and Liou et al. demonstrated that fluoroquinolone resistance correlated with fluoroquinolone consumption in Europe and Taiwan, respectively. Additionally, dietary factors such as high consumption of fermented foods or probiotic yogurts, which can influence gastric microbiota composition, may also play a role in modulating bacterial adaptation or survival and be associated with the development of resistance or increased/reduced risk of infection.”

  1. The study suggests co-infection with wild-type and mutant strains occurs but does not explore potential mechanisms (e.g., reinfection vs. co-colonization).

Thank you for your valuable comment. We agree that understanding the mechanisms behind co-infection with wild-type and mutant strains, such as reinfection versus co-colonization, is crucial. We have added and developed explanations in the discussion to expand the possible causes about this issue (in lines 227-247):

„We have not investigated the specific mechanisms underlying this phenomenon. However, two main possibilities should be considered - reinfection with a different strain or co-colonization by both wild-type and mutant strains. Reinfection refers to the acquisition of a new strain of H. pylori, possibly with different resistance patterns, after the initial infection has been treated or resolved. Co-colonization occurs when multiple strains exist simultaneously within the same host. In the case of H. pylori, this could lead to complex interactions between wild-type and resistant strains, potentially influencing treatment outcomes and the development of resistance. Previous studies have shown that co-colonization with different H. pylori strains may have important implications for the persistence of infection and the success of eradication therapy and should be considered when interpreting the results of antimicrobial susceptibility tests. It has been suggested that co-colonization could lead to competitive interactions between strains, potentially influencing the overall resistance profile of the bacterial population within the host. Such dynamics may contribute to the failure of standard treatment regimens and force a more personalized approach to therapy. Distinguishing between reinfection and co-colonization can be challenging, especially without detailed molecular analysis. Techniques such as strain typing and next-generation sequencing or RAPD fingerprinting could help clarify these mechanisms and provide more precise insights into the role of strain diversity in resistance development. We will consider their use in future studies to obtain a complete picture of the molecular mechanisms responsible for resistance”.

  1. A brief mention of potential clinical implications for treatment failure and alternative antibiotic strategies (e.g., tailored therapy approaches) would strengthen the discussion.

We have added a section in the discussion focus on these issues to enhance the discussion (in lines 309-332):

„A significant clinical concern associated with the increasing prevalence of H. pylori resistance to clarithromycin is the potential for treatment failure. As resistance rates rise, standard therapy may no longer be effective, leading to persistent infections and complications such as peptic ulcers or gastric cancer. In such cases, tailored therapy approaches, including the use of alternative antibiotics or combination therapies, are essential to improve treatment outcomes. In the metagenomic analysis, Ma et al. was shown that in first-line treatment, tailored therapy proved to be more effective than empirical therapy. In second-line therapy, no significant differences were observed between the two approaches. A similar outcome was found in treatments involving a combination of second- and third-line therapies. Regarding adverse events, no major differences were noted between the two treatment options. In the subgroup analysis of various empirical approaches, tailored therapy demonstrated the most significant benefit compared to the conventional triple therapy.

In turn, in study conducted by Kim et al., demonstrated that tailored therapy based on genotypic resistance was significantly more effective than empirical therapy for H. pylori eradication. The eradication rates were 65.7% (201/306) in the empirical therapy group and 81.9% (235/287) in the tailored therapy group. There was no difference in compliance between the two groups. In a meta-analysis conducted by Rokkas et al. [11], the researchers showed that although tailored therapy proved to be more effective than empirical treatment, a >90% eradication rate was achieved in only 15 (44%) of the studies, and >95% in only 6 (17.6%). Based on this, the researchers concluded that while tailored therapy was superior to empirical treatment, the lack of therapy optimization prevented the achievement of high cure rates (>90%). These results highlight that H. pylori infection, like other infectious diseases, should follow the principles of antimicrobial stewardship in line with treatment guidelines”.

Reviewer 2 Report

Comments and Suggestions for Authors

This manuscript shows data from a study aimed to assess the prevalence and variability of clarithromycin resistance mutations in pediatric patients from Bydgoszcz, Poland.

The design of the study is correct but very simple. The number of samples (45) does not correspond with the number of patients (40). Figure 1, 2 and table 2 are redundant and they must be combined in a single figure, and table 2 could be included as supplementary materials.

Clarithromycin resistance patterns of all the isolates must be included. Even of those, where DNA was not detected, or it was an invalid twice. Moreover, data must be discussed compared with the actual mutation found.
The discussion is well documented, it provides data of different studies around the world, and final conclusion is sound.
No data are shown about the actual pathology of the patients. Due to the pediatric nature of the samples, it is not expected gastric cancer, but the nature of the pathology based on the gastroscopic/histopathological examination must be provided and discussed.

Author Response

Dear Reviewer,

We thank you for your valuable comments to our manuscript. We carefully considered your comments and revised the paper according your suggestions. We responded to your comments below.

The number of samples (45) does not correspond with the number of patients (40).

Thank you for your valuable comment. We sincerely apologize for the mistake. The correct number of patients is 45, and we have corrected this number in the manuscript.

Figure 1, 2 and table 2 are redundant and they must be combined in a single figure, and table 2 could be included as supplementary materials.

We have improved the figures, and the table has been moved to the supplementary material.

Clarithromycin resistance patterns of all the isolates must be included. Even of those, where DNA was not detected, or it was an invalid twice. Moreover, data must be discussed compared with the actual mutation found.

Thank you very much for these comments. In the following lines: 266-289, we have added the text below to enrich the discussion, ensuring that all our obtained results are taken into account.

„In our study, two samples were negative for H. pylori DNA despite positive histopathology and gastroscopy results, while one sample gave an „invalid” result. These discrepancies highlight the potential challenges of molecular diagnostics. One possible reason for the failure to detect H. pylori DNA is a low bacterial load in the sample, which may affect the sensitivity of the genotyping test. In addition, the presence of PCR inhibitors such as mucus, blood components, or other substances may interfere with the amplification process, leading to false negative results. Another factor to consider is the inherent methodological differences between diagnostic techniques: while histopathology and gastroscopy identify H. pylori at the tissue level, molecular tests rely on the presence of bacterial DNA, which cannot always be adequately extracted or detected. We cannot exclude that the mentioned obstacles caused an error during DNA isolation of the genetic material.

The „invalid” result observed in one sample, despite repeated testing, raises concerns about potential technical issues; however, similar factors as in the case of a negative result may also contribute to this outcome, such as: an insufficient amount of genetic material, or the presence of substances inhibiting the PCR. Such incidents highlight the importance of proper sample collection, handling and processing to minimize potential errors. These results emphasize the need for a comprehensive diagnostic approach to detect H. pylori, especially in pediatric patients. While molecular tests provide valuable information on resistance mutations, detection should be based on different tests to verify the obtained results. It seems that combining genotypic methods with traditional histopathological and endoscopic techniques can increase diagnostic accuracy and improve patient management. In cases where molecular tests give equivocal or negative results despite clinical suspicion, consideration should be given to repeating the test on another sample or using an alternative accessible diagnostic method.”

No data are shown about the actual pathology of the patients. Due to the pediatric nature of the samples, it is not expected gastric cancer, but the nature of the pathology based on the gastroscopic/histopathological examination must be provided and discussed.

We have included the histopathological data in the statistical analysis and added the relevant sections to the Methods and Discussion in the following lines: 131-139 and 290-308. Below, we attach the text introduced in the given lines:

„To assess the association between H. pylori strain type (WT or with mutation) and both the intensity of inflammation activity grade and colonization density grade, statistical analysis was performed. Additionally, the relationship between strain type and the Ct value obtained during qPCR was analyzed, as Ct is inversely related to the amount of DNA present in the sample. No significant association was observed between the presence/absence of mutations and inflammation grade or colonization density grade (p = 0.868; p = 0.818, respectively). However, statistical analysis revealed that wild-type strains (without mutations) had a higher Ct value (which indicates a lower amount of DNA present in the sample) compared to mutated strains (p = 0.014)”

„Interestingly, statistical analysis showed that Ct values obtained for the samples with mutated H. pylori strains presence were lower, which is equivalent to a higher amount of H. pylori DNA in the sample compared to wild-type strains. This difference in Ct values may be explained by several factors. Mutated strains may show increased colonization ability or increased resistance to treatment, which could lead to a higher bacterial load in the sample. Resistance to certain antibiotics, enabling them to colonize and multiply in the stomach environment and potentially increase their abundance in the sample, thus contributing to a higher overall H. pylori DNA concentration. Moreover, genetic changes in H. pylori could impact its pathogenicity, leading to a more intense infection. This increased pathogenicity could result in a higher bacterial load in the sample, which would be reflected by a lower Ct value. Therefore, the observed difference in Ct values between the WT and mutant strains may provide valuable insights into the underlying infection dynamics and the bacterial load associated with different H. pylori genotypes. The lack of statistically significant associations between H. pylori strain type (WT or mutant) and both inflammation grade and colonization density grade suggests that the presence of mutations associated with antibiotic resistance does not significantly impact on the histopathological outcome. This indicate that factors such as the host's immune response, bacterial virulence mechanisms unrelated to antibiotic resistance, infection duration, and environmental factors have a greater influence on gastric mucosal alterations”

Round 2

Reviewer 2 Report

Comments and Suggestions for Authors

The authors have addressed correctly all the requested changes and comments.